# A Finite-Particle Convergence Rate for
# Stein Variational Gradient Descent

**Jiaxin Shi**[*]
Stanford University
Stanford, CA 94305
jiaxins@stanford.edu

**Lester Mackey**
Microsoft Research New England
Cambridge, MA 02474
lmackey@microsoft.com

## Abstract

We provide the first finite-particle convergence rate for Stein variational gradient descent (SVGD), a popular algorithm for approximating a probability distribution with a collection of particles. Specifically, whenever the target distribution is sub-Gaussian with a Lipschitz score, SVGD with $n$ particles and an appropriate step size sequence drives the kernel Stein discrepancy to zero at an order $1/\sqrt{\log \log n}$ rate. We suspect that the dependence on $n$ can be improved, and we hope that our explicit, non-asymptotic proof strategy will serve as a template for future refinements.

## 1  Introduction

Stein variational gradient descent [SVGD, 18] is an algorithm for approximating a target probability distribution $P$ on $\mathbb{R}^d$ with a collection of $n$ particles. Given an initial particle approximation $\mu_0^n = \frac{1}{n}\sum_{i=1}^n \delta_{x_i}$ with locations $x_i \in \mathbb{R}^d$, SVGD (Algorithm 1) iteratively evolves the particle locations to provide a more faithful approximation of the target $P$ by performing optimization in the space of probability measures. SVGD has demonstrated encouraging results for a wide variety of inferential tasks, including approximate inference [18, 28, 27], generative modeling [26, 13], and reinforcement learning [11, 21].

Despite the popularity of SVGD, relatively little is known about its approximation quality. A first analysis by Liu [17, Thm. 3.3] showed that *continuous SVGD*—that is, Algorithm 2 initialized with a continuous distribution $\mu_0^\infty$ in place of the discrete particle approximation $\mu_0^n$—converges to $P$ in kernel Stein discrepancy [KSD, 4, 19, 9]. KSD convergence is also known to imply weak convergence [9, 3, 12, 1] and Wasserstein convergence [14] under various conditions on the target $P$ and the SVGD kernel $k$. Follow-up work by Korba et al. [15], Salim et al. [22], Sun et al. [24] sharpened the result of Liu with path-independent constants, weaker smoothness conditions, and explicit rates of convergence. In addition, Duncan et al. [6] analyzed the continuous-time limit of continuous SVGD to provide conditions for exponential convergence. However, each of these analyses applies only to continuous SVGD and not to the finite-particle algorithm used in practice.

To bridge this gap, Liu [17, Thm. 3.2] showed that $n$-particle SVGD converges to continuous SVGD in bounded-Lipschitz distance but only under boundedness assumptions violated by most applications of SVGD. To provide a more broadly applicable proof of convergence, Gorham et al. [10, Thm. 7] showed that $n$-particle SVGD converges to continuous SVGD in 1-Wasserstein distance under assumptions commonly satisfied in SVGD applications. However, both convergence results are asymptotic, providing neither explicit error bounds nor rates of convergence. Korba et al. [15, Prop. 7] explicitly bounded the expected squared Wasserstein distance between $n$-particle and continuous SVGD but only under the assumption of bounded $\nabla \log p$, an assumption that rules out all strongly

---

[*]Part of this work was done at Microsoft Research New England.

37th Conference on Neural Information Processing Systems (NeurIPS 2023).

---

**Algorithm 1** $n$-particle Stein Variational Gradient Descent [18]: $\text{SVGD}(\mu_0^n, r)$

---

**Input:** Target $P$ with density $p$, kernel $k$, step sizes $(\epsilon_s)_{s \geq 0}$, particle approximation $\mu_0^n = \frac{1}{n}\sum_{i=1}^n \delta_{x_i}$, rounds $r$
**for** $s = 0, \cdots, r-1$ **do**
    $x_i \leftarrow x_i + \epsilon_s \frac{1}{n}\sum_{j=1}^n k(x_j, x_i)\nabla \log p(x_j) + \nabla_x k(x_j, x_i)$ for $i = 1, \ldots, n$.
**Output:** Updated approximation $\mu_r^n = \frac{1}{n}\sum_{i=1}^n \delta_{x_i}$ of the target $P$

---

---

**Algorithm 2** Generic Stein Variational Gradient Descent [18]: $\text{SVGD}(\mu_0, r)$

---

**Input:** Target $P$ with density $p$, kernel $k$, step sizes $(\epsilon_s)_{s \geq 0}$, approximating measure $\mu_0$, rounds $r$
**for** $s = 0, \cdots, r-1$ **do**
    Let $\mu_{s+1}$ be the distribution of $X^s + \epsilon_s \int k(x, X^s)\nabla \log p(x) + \nabla_x k(x, X^s)d\mu_s(x)$ for $X^s \sim \mu_s$.
**Output:** Updated approximation $\mu_r$ of the target $P$

---

log concave or dissipative distributions and all distributions for which the KSD is currently known to control weak convergence [9, 3, 12, 1]. In addition, Korba et al. [15] do not provide a unified bound for the convergence of $n$-particle SVGD to $P$ and ultimately conclude that "the convergence rate for SVGD using $[\mu_0^n]$ remains an open problem." The same open problem was underscored in the later work of Salim et al. [22], who write "an important and difficult open problem in the analysis of SVGD is to characterize its complexity with a finite number of particles."

In this work, we derive the first unified convergence bound for finite-particle SVGD to its target. To achieve this, we first bound the 1-Wasserstein discretization error between finite-particle and continuous SVGD under assumptions commonly satisfied in SVGD applications and compatible with KSD weak convergence control (see Theorem 1). We next bound KSD in terms of 1-Wasserstein distance and SVGD moment growth to explicitly control KSD discretization error in Theorem 2. Finally, Theorem 3 combines our results with the established KSD analysis of continuous SVGD to arrive at an explicit KSD error bound for $n$-particle SVGD.

## 2 Notation and Assumptions

Throughout, we fix a nonnegative step size sequence $(\epsilon_s)_{s \geq 0}$, a target distribution $P$ in the set $\mathcal{P}_1$ of probability measures on $\mathbb{R}^d$ with integrable first moments, and a *reproducing kernel* $k$—a symmetric positive definite function mapping $\mathbb{R}^d \times \mathbb{R}^d$ to $\mathbb{R}$—with reproducing kernel Hilbert space (RKHS) $\mathcal{H}$ and product RKHS $\mathcal{H}^d \triangleq \bigotimes_{i=1}^d \mathcal{H}$ [2]. We will use the terms "kernel" and "reproducing kernel" interchangeably. For all $\mu, \nu \in \mathcal{P}_1$, we let $\Gamma(\mu, \nu)$ be the set of all *couplings* of $\mu$ and $\nu$, i.e., joint probability distributions $\gamma$ over $\mathcal{P}_1 \times \mathcal{P}_1$ with $\mu$ and $\nu$ as marginal distributions of the first and second variable respectively. We further let $\mu \otimes \nu$ denote the independent coupling, the distribution of $(X, Z)$ when $X$ and $Z$ are drawn independently from $\mu$ and $\nu$ respectively. With this notation in place we define the 1-Wasserstein distance between $\mu, \nu \in \mathcal{P}_1$ as $W_1(\mu, \nu) \triangleq \inf_{\gamma \in \Gamma(\mu,\nu)} \mathbb{E}_{(X,Z) \sim \gamma}[\|Z - X\|_2]$ and introduce the shorthand $m_{\mu, x^\star} \triangleq \mathbb{E}_\mu[\|\cdot - x^\star\|]_2$ for each $x^\star \in \mathbb{R}^d$, $m_{\mu, P} \triangleq \mathbb{E}_{(X,Z) \sim \mu \otimes P}[\|X - Z\|_2]$, and $M_{\mu, P} \triangleq \mathbb{E}_{(X,Z) \sim \mu \otimes P}[\|X - Z\|_2^2]$. We further define the Kullback-Leibler (KL) divergence as $\text{KL}(\mu \| \nu) \triangleq \mathbb{E}_\mu[\log(\frac{d\mu}{d\nu})]$ when $\mu$ is absolutely continuous with respect to $\nu$ (denoted by $\mu \ll \nu$) and as $\infty$ otherwise.

Our analysis will make use of the following assumptions on the SVGD kernel and target distribution.

**Assumption 1** (Lipschitz, mean-zero score function)**.** *The target distribution $P \in \mathcal{P}_1$ has a differentiable density $p$ with an $L$-Lipschitz score function $s_p \triangleq \nabla \log p$, i.e., $\|s_p(x) - s_p(y)\|_2 \leq L\|x - y\|_2$ for all $x, y \in \mathbb{R}^d$. Moreover, $\mathbb{E}_P[s_p] = 0$ and $s_p(x^\star) = 0$ for some $x^\star \in \mathbb{R}^d$.*

**Assumption 2** (Bounded kernel derivatives)**.** *The kernel $k$ is twice differentiable and $\sup_{x,y \in \mathbb{R}^d} \max(|k(x,y)|, \|\nabla_x k(x,y)\|_2, \|\nabla_y \nabla_x k(x,y)\|_{\text{op}}, \|\nabla_x^2 k(x,y)\|_{\text{op}}) \leq \kappa_1^2$ for $\kappa_1 > 0$. Moreover, for all $i, j \in \{1, 2, \ldots, d\}$, $\sup_{x \in \mathbb{R}^d} \nabla_{y_i} \nabla_{y_j} \nabla_{x_i} \nabla_{x_j} k(x,y)|_{y=x} \leq \kappa_2^2$ for $\kappa_2 > 0$.*

**Assumption 3** (Decaying kernel derivatives)**.** *The kernel $k$ is differentiable and admits a $\gamma \in \mathbb{R}$ such that, for all $x, y \in \mathbb{R}^d$ satisfying $\|x - y\|_2 \geq 1$,*

$$\|\nabla_x k(x,y)\|_2 \leq \gamma/\|x - y\|_2.$$

Assumptions 1, 2, and 3 are both commonly invoked and commonly satisfied in the literature. For example, the Lipschitz score assumption is consistent with prior SVGD convergence analyses [17, 15, 22] and, by Gorham and Mackey [8, Prop. 1], the score $s_p$ is mean-zero under the mild integrability condition $\mathbb{E}_{X \sim P}[\|s_p(X)\|_2] < \infty$. The bounded and decaying derivative assumptions have also been made in prior analyses [15, 9] and, as we detail in Appendix A, are satisfied by the kernels most commonly used in SVGD, like the Gaussian and inverse multiquadric (IMQ) kernels. Notably, in these cases, the bounds $\kappa_1$ and $\kappa_2$ are independent of the dimension $d$.

To leverage the continuous SVGD convergence rates of Salim et al. [22], we additionally assume that the target $P$ satisfies Talagrand's $T_1$ inequality [25, Def. 22.1]. Remarkably, Villani [25, Thm. 22.10] showed that Assumption 4 is *equivalent* to $P$ being a sub-Gaussian distribution. Hence, this mild assumption holds for all strongly log concave $P$ [23, Def. 2.9], all $P$ satisfying the log Sobolev inequality [25, Thm. 22.17], and all *distantly dissipative* $P$ for which KSD is known to control weak convergence [9, Def. 4].

**Assumption 4** (Talagrand's $T_1$ inequality [25, Def. 22.1]). *For $P \in \mathcal{P}_1$, there exists $\lambda > 0$ such that, for all $\mu \in \mathcal{P}_1$,*

$$W_1(\mu, P) \le \sqrt{2\mathrm{KL}(\mu\|P)/\lambda}.$$

Finally we make use of the following notation specific to the SVGD algorithm.

**Definition 1** (Stein operator). *For any differentiable vector-valued function $g : \mathbb{R}^d \to \mathbb{R}^d$, the* Langevin Stein operator [8] *for $P$ satisfying Assumption 1 is defined by*

$$(\mathcal{T}_P g)(x) \triangleq \langle s_p(x), g(x) \rangle + \nabla \cdot g(x) \quad \text{for all} \quad x \in \mathbb{R}^d.$$

**Definition 2** (Vector-valued Stein operator). *For any differentiable function $h : \mathbb{R}^d \to \mathbb{R}$, the* vector-valued Langevin Stein operator [19] *for $P$ satisfying Assumption 1 is defined by*

$$(\mathcal{A}_P h)(x) \triangleq s_p(x)h(x) + \nabla h(x) \quad \text{for all} \quad x \in \mathbb{R}^d.$$

**Definition 3** (SVGD transport map and pushforward). *The* SVGD transport map [18] *for a target $P$ satisfying Assumption 1, a kernel $k$ satisfying Assumption 2, a step size $\epsilon \ge 0$, and an approximating distribution $\mu \in \mathcal{P}_1$ takes the form*

$$T_{\mu,\epsilon}(x) \triangleq x + \epsilon \mathbb{E}_{X \sim \mu}[(\mathcal{A}_P k(\cdot, x))(X)] \quad \text{for all} \quad x \in \mathbb{R}^d.$$

*Moreover, the* SVGD pushforward $\Phi_\epsilon(\mu)$ *represents the distribution of $T_{\mu,\epsilon}(X)$ when $X \sim \mu$.*

**Definition 4** (Kernel Stein discrepancy). *The* Langevin kernel Stein discrepancy [KSD, 4, 19, 9] *for $P$ satisfying Assumption 1, $k$ satisfying Assumption 2, and measures $\mu, \nu \in \mathcal{P}_1$ is given by*

$$\mathrm{KSD}_P(\mu, \nu) \triangleq \sup_{\|g\|_{\mathcal{H}^d} \le 1} \mathbb{E}_\mu[\mathcal{T}_P g] - \mathbb{E}_\nu[\mathcal{T}_P g].$$

Notably, the KSD so-defined is symmetric in its two arguments and satisfies the triangle inequality.

**Lemma 1** (KSD symmetry and triangle inequality). *Under Definition 4, for all $\mu, \nu, \pi \in \mathcal{P}_1$,*

$$\mathrm{KSD}_P(\mu, \nu) = \mathrm{KSD}_P(\nu, \mu) \quad and \quad \mathrm{KSD}_P(\mu, \nu) \le \mathrm{KSD}_P(\mu, \pi) + \mathrm{KSD}_P(\pi, \nu).$$

*Proof.* Fix any $\mu, \nu, \pi \in \mathcal{P}_1$. For symmetry, we note that $g \in \mathcal{H}^d \Leftrightarrow f = -g \in \mathcal{H}^d$, so

$$\mathrm{KSD}_P(\mu, \pi) = \sup_{\|g\|_{\mathcal{H}^d} \le 1} \mathbb{E}_\mu[\mathcal{T}_P g] - \mathbb{E}_\pi[\mathcal{T}_P g] = \sup_{\|f\|_{\mathcal{H}^d} \le 1} \mathbb{E}_\pi[\mathcal{T}_P f] - \mathbb{E}_\mu[\mathcal{T}_P f] = \mathrm{KSD}_P(\pi, \mu).$$

To establish the triangle inequality, we write

$$\begin{aligned}
\mathrm{KSD}_P(\mu, \nu) &= \sup_{\|g\|_{\mathcal{H}^d} \le 1} \mathbb{E}_\mu[\mathcal{T}_P g] - \mathbb{E}_\pi[\mathcal{T}_P g] + \mathbb{E}_\pi[\mathcal{T}_P g] - \mathbb{E}_\nu[\mathcal{T}_P g] \\
&\le \sup_{\|g\|_{\mathcal{H}^d} \le 1} (\mathbb{E}_\mu[\mathcal{T}_P g] - \mathbb{E}_\pi[\mathcal{T}_P g]) + \sup_{\|h\|_{\mathcal{H}^d} \le 1} (\mathbb{E}_\pi[\mathcal{T}_P h] - \mathbb{E}_\nu[\mathcal{T}_P h]) \\
&\le \mathrm{KSD}_P(\mu, \pi) + \mathrm{KSD}_P(\pi, \nu).
\end{aligned}$$

$\square$

# 3  Wasserstein Discretization Error of SVGD

Our first main result concerns the discretization error of SVGD and shows that $n$-particle SVGD remains close to its continuous SVGD limit whenever the step size sum $b_{r-1} = \sum_{s=0}^{r-1} \epsilon_s$ is sufficiently small.

**Theorem 1** (Wasserstein discretization error of SVGD). *Suppose Assumptions 1, 2, and 3 hold. For any $\mu_0^n, \mu_0^\infty \in \mathcal{P}_1$, the outputs $\mu_r^n = \mathrm{SVGD}(\mu_0^n, r)$ and $\mu_r^\infty = \mathrm{SVGD}(\mu_0^\infty, r)$ of Algorithm 2 satisfy*

$$\log\left(\frac{W_1(\mu_r^n, \mu_r^\infty)}{W_1(\mu_0^n, \mu_0^\infty)}\right) \le b_{r-1}(A + B\exp(Cb_{r-1}))$$

*for $b_{r-1} \triangleq \sum_{s=0}^{r-1} \epsilon_s$, $A = (c_1 + c_2)(1 + m_{P,x^\star})$, $B = c_1 m_{\mu_0^n, P} + c_2 m_{\mu_0^\infty, P}$, $C = \kappa_1^2(3L + 1)$, $c_1 = \max(\kappa_1^2 L, \kappa_1^2)$, and $c_2 = \kappa_1^2(L+1) + L\max(\gamma, \kappa_1^2)$.*

We highlight that Theorem 1 applies to **any** $\mathcal{P}_1$ initialization of SVGD: the initial particles supporting $\mu_0^n$ could, for example, be drawn i.i.d. from a convenient auxiliary distribution $\mu_0^\infty$ or even generated deterministically from some quadrature rule. To marry this result with the continuous SVGD convergence bound of Section 5, we will ultimately require $\mu_0^\infty$ to be a continuous distribution with finite $\mathrm{KL}(\mu_0^\infty \| P)$. Hence, our primary desideratum for SVGD initialization is that $\mu_0^n$ have small Wasserstein distance to some $\mu_0^\infty$ with $\mathrm{KL}(\mu_0^\infty \| P) < \infty$. Then, by Theorem 1, the SVGD discretization error $W_1(\mu_r^n, \mu_r^\infty)$ will remain small whenever the step size sum is not too large.

The proof of Theorem 1 in Section 6 relies on two lemmas. The first, due to Gorham et al. [10], shows that the one-step SVGD pushforward map $\Phi_\epsilon$ (Definition 3) is pseudo-Lipschitz with respect to the 1-Wasserstein distance[2] whenever the score function $\nabla \log p$ and kernel $k$ fulfill a commonly-satisfied pseudo-Lipschitz condition. Here, for any $g : \mathbb{R}^d \to \mathbb{R}^d$, we define the Lipschitz constant $\mathrm{Lip}(g) \triangleq \sup_{x,z \in \mathbb{R}^d} \|g(x) - g(z)\|_2 / \|x - z\|_2$.

**Lemma 2** (Wasserstein pseudo-Lipschitzness of SVGD [10, Lem. 12]). *For $P$ satisfying Assumption 1, suppose that the following pseudo-Lipschitz bounds hold*

$$\mathrm{Lip}(s_p(x)k(x, \cdot) + \nabla_x k(x, \cdot)) \le c_1(1 + \|x - x^\star\|_2),$$
$$\mathrm{Lip}(s_p k(\cdot, z) + \nabla_x k(\cdot, z)) \le c_2(1 + \|z - x^\star\|_2).$$

*for some constants $c_1, c_2 \in \mathbb{R}$ and all $x, z \in \mathbb{R}^d$. Then, for any $\mu, \nu \in \mathcal{P}_1$,*

$$W_1(\Phi_\epsilon(\mu), \Phi_\epsilon(\nu)) \le W_1(\mu, \nu)(1 + \epsilon c_{\mu,\nu}),$$

*where $\Phi_\epsilon$ is the one-step SVGD pushforward (Definition 3) and $c_{\mu,\nu} = c_1(1 + m_{\mu,x^\star}) + c_2(1 + m_{\nu,x^\star})$.*

In Section 6, we will show that, under Assumptions 1, 2, and 3, the preconditions of Lemma 2 are fulfilled with $c_1$ and $c_2$ exactly as in Theorem 1. The second lemma, proved in Section 7, controls the growth of the first and second absolute moments under SVGD.

**Lemma 3** (SVGD moment growth). *Suppose Assumptions 1 and 2 hold, and let $C = \kappa_1^2(3L + 1)$. Then the SVGD output $\mu_r$ of Algorithm 2 with $b_{r-1} \triangleq \sum_{s=0}^{r-1} \epsilon_s$ satisfies*

$$m_{\mu_r, x^\star} - m_{P, x^\star} \le m_{\mu_r, P} \le m_{\mu_0, P} \prod_{s=0}^{r-1}(1 + \epsilon_s C) \le m_{\mu_0, P}\exp(Cb_{r-1}),$$
$$M_{\mu_r, P} \le M_{\mu_0, P} \prod_{s=0}^{r-1}(1 + \epsilon_s C)^2 \le M_{\mu_0, P}\exp(2Cb_{r-1}).$$

The key to the proof of Lemma 3 is that we show the norm of any SVGD update, i.e., $\|T_{\mu,\epsilon}(x) - x\|_2$, is controlled by $m_{\mu, P}$, the first absolute moment of $\mu$ measured against $P$. This is mainly due to the Lipschitzness of the score function $s_p$ and our assumptions on the boundedness of the kernel and its derivatives. Then, we can use the result to control the growth of $m_{\mu_r, P}$ across iterations since $m_{\mu_{r+1}, P} = \mathbb{E}_{(X,Z) \sim \mu_r \otimes P}[\|T_{\mu_r, \epsilon_r}(X) - Z\|_2]$. The same strategy applies to the second absolute moment $M_{\mu, P}$. The proof of Theorem 1 then follows directly from Lemma 2 where we plug in the first moment bound of Lemma 3.

---

[2] We say a map $\Phi : \mathcal{P}_1 \to \mathcal{P}_1$ is *pseudo-Lipschitz* with respect to 1-Wasserstein distance if, for a constant $C_\Phi \in \mathbb{R}$, some $x^\star \in \mathbb{R}^d$, and all $\mu, \nu \in \mathcal{P}_1$, $W_1(\Phi(\mu), \Phi(\nu)) \le W_1(\mu, \nu)(1 + m_{\mu, x^\star} + m_{\nu, x^\star})C_\Phi$.

# 4 KSD Discretization Error of SVGD

Our next result translates the Wasserstein error bounds of Theorem 1 into KSD error bounds.

**Theorem 2** (KSD discretization error of SVGD). *Suppose Assumptions 1, 2, and 3 hold. For any $\mu_0^n, \mu_0^\infty \in \mathcal{P}_1$, the outputs of Algorithm 2, $\mu_r^n = \mathrm{SVGD}(\mu_0^n, r)$ and $\mu_r^\infty = \mathrm{SVGD}(\mu_0^\infty, r)$, satisfy*

$$\mathrm{KSD}_P(\mu_r^n, \mu_r^\infty) \le (\kappa_1 L + \kappa_2 d) w_{0,n} \exp(b_{r-1}(A + B \exp(C b_{r-1})))$$
$$+ \kappa_1 d^{1/4} L \sqrt{2 M_{\mu_0^\infty, P} w_{0,n}} \exp(b_{r-1}(2C + A + B \exp(C b_{r-1}))/2)$$

*for $w_{0,n} \triangleq W_1(\mu_0^n, \mu_0^\infty)$ and $A, B, C$ defined as in Theorem 1.*

Our proof of Theorem 2 relies on the following lemma, proved in Section 8, that shows that the KSD is controlled by the 1-Wasserstein distance.

**Lemma 4** (KSD-Wasserstein bound). *Suppose Assumptions 1 and 2 hold. For any $\mu, \nu \in \mathcal{P}_1$,*

$$\mathrm{KSD}_P(\mu, \nu) \le (\kappa_1 L + \kappa_2 d) W_1(\mu, \nu) + \kappa_1 d^{1/4} L \sqrt{2 M_{\nu, P} W_1(\mu, \nu)}.$$

Lemma 4 is proved in two steps. We first linearize $(\mathcal{T}_P g)(x)$ in the KSD definition through the Lipschitzness of $s_p$ and the boundedness and Lipschitzness of RKHS functions. Then, we assign a 1-Wasserstein optimal coupling of $(\mu, \nu)$ to obtain the Wasserstein bound on the right.

**Proof of Theorem 2** The result follows directly from Lemma 4, the second moment bound of Lemma 3, and Theorem 1. ∎

# 5 A Finite-particle Convergence Rate for SVGD

To establish our main SVGD convergence result, we combine Theorems 1 and 2 with the following descent lemma for continuous SVGD error due to Salim et al. [22] which shows that continuous SVGD decreases the KL divergence to $P$ and drives the KSD to $P$ to zero.

**Lemma 5** (Continuous SVGD descent lemma [22, Thm. 3.2]). *Suppose Assumptions 1, 2, and 4 hold, and consider the outputs $\mu_r^\infty = \mathrm{SVGD}(\mu_0^\infty, r)$ and $\mu_{r+1}^\infty = \mathrm{SVGD}(\mu_0^\infty, r+1)$ of Algorithm 2 with $\mu_0^\infty \ll P$. If $\max_{0 \le s \le r} \epsilon_s \le R_{\alpha,2}$ for some $\alpha > 1$ and*

$$R_{\alpha,p} \triangleq \min\left(\frac{p}{\kappa_1^2(L+\alpha^2)}, (\alpha-1)\left(1 + L m_{\mu_0^\infty, x^\star} + 2L\sqrt{\frac{2}{\lambda} \mathrm{KL}(\mu_0^\infty \| P)}\right)\right) \text{ for } p \in \{1, 2\}, \quad (1)$$

*then*

$$\mathrm{KL}(\mu_{r+1}^\infty \| P) - \mathrm{KL}(\mu_r^\infty \| P) \le -\epsilon_r \left(1 - \frac{\kappa_1^2(L+\alpha^2)}{2} \epsilon_r\right) \mathrm{KSD}_P(\mu_r^\infty, P)^2. \quad (2)$$

By summing the result (2) over $r \in \{0, \dots, t\}$, we obtain the following corollary.

**Corollary 1.** *Under the assumptions and notation of Lemma 5, suppose $\max_{0 \le r \le t} \epsilon_r \le R_{\alpha,1}$ for some $\alpha > 1$, and let $\pi_r \triangleq \frac{c(\epsilon_r)}{\sum_{r=0}^{t} c(\epsilon_r)}$ for $c(\epsilon) \triangleq \epsilon\left(1 - \frac{\kappa_1^2(L+\alpha^2)}{2}\epsilon\right)$. Since $\frac{\epsilon}{2} \le c(\epsilon) < \epsilon$, we have*

$$\sum_{r=0}^{t} \pi_r \mathrm{KSD}_P(\mu_r^\infty, P)^2 \le \frac{1}{\sum_{r=0}^{t} c(\epsilon_r)} \mathrm{KL}(\mu_0^\infty \| P) \le \frac{2}{\sum_{r=0}^{t} \epsilon_r} \mathrm{KL}(\mu_0^\infty \| P).$$

Finally, we arrive at our main result that bounds the approximation error of $n$-particle SVGD in terms of the chosen step size sequence and the initial discretization error $W_1(\mu_0^n, \mu_0^\infty)$.

**Theorem 3** (KSD error of finite-particle SVGD). *Suppose Assumptions 1, 2, 3, and 4 hold, fix any $\mu_0^\infty \ll P$ and $\mu_0^n \in \mathcal{P}_1$, and let $w_{0,n} \triangleq W_1(\mu_0^n, \mu_0^\infty)$. If $\max_{0 \le r < t} \epsilon_r \le \epsilon_t \triangleq R_{\alpha,1}$[3] for some $\alpha > 1$ and $R_{\alpha,1}$ defined in Lemma 5, then the Algorithm 2 outputs $\mu_r^n = \mathrm{SVGD}(\mu_0^n, r)$ satisfy*

$$\min_{0 \le r \le t} \mathrm{KSD}_P(\mu_r^n, P) \le \sum_{r=0}^{t} \pi_r \mathrm{KSD}_P(\mu_r^n, P) \le a_{t-1} + \sqrt{\frac{2}{R_{\alpha,1}+b_{t-1}} \mathrm{KL}(\mu_0^\infty \| P)}, \quad (3)$$

*for $\pi_r$ as defined in Lemma 5, $(A, B, C)$ as defined in Theorem 1, $b_{t-1} \triangleq \sum_{r=0}^{t-1} \epsilon_r$, and*

$$a_{t-1} \triangleq (\kappa_1 L + \kappa_2 d) w_{0,n} \exp(b_{t-1}(A + B \exp(C b_{t-1}))) \quad (4)$$
$$+ \kappa_1 d^{1/4} L \sqrt{2 M_{\mu_0^\infty, P} w_{0,n}} \exp(b_{t-1}(2C + A + B \exp(C b_{t-1}))/2).$$

---

[3] Note that the value assigned to $\epsilon_t$ does not have any impact on the algorithm that generates $\mu_r^n$ when $r \le t$.

*Proof.* By the triangle inequality (Lemma 1) and Theorem 2 we have

$$|\text{KSD}_P(\mu_r^n, P) - \text{KSD}_P(\mu_r^\infty, P)| \leq \text{KSD}_P(\mu_r^n, \mu_r^\infty) \leq a_{r-1}$$

for each $r$. Therefore

$$\sum_{r=0}^t \pi_r (\text{KSD}_P(\mu_r^n, P) - a_{r-1})^2 \leq \sum_{r=0}^t \pi_r \text{KSD}_P(\mu_r^\infty, P)^2 \leq \frac{2}{R_{\alpha,1} + b_{t-1}} \text{KL}(\mu_0^\infty \| P), \quad (5)$$

where the last inequality follows from Corollary 1. Moreover, by Jensen's inequality,

$$\sum_{r=0}^t \pi_r (\text{KSD}_P(\mu_r^n, P) - a_{r-1})^2 \geq \left( \sum_{r=0}^t \pi_r \text{KSD}_P(\mu_r^n, P) - \sum_{r=0}^t \pi_r a_{r-1} \right)^2. \quad (6)$$

Combining (5) and (6), we have

$$\sum_{r=0}^t \pi_r \text{KSD}_P(\mu_r^n, P) \leq \sum_{r=0}^t \pi_r a_{r-1} + \sqrt{\frac{2}{R_{\alpha,1} + b_{t-1}} \text{KL}(\mu_0^\infty \| P)}.$$

We finish the proof by noticing that $\sum_{r=0}^t \pi_r a_{r-1} \leq \max_{0 \leq r \leq t} a_{r-1} = a_{t-1}$. $\qquad \square$

The following corollary, proved in Appendix B, provides an explicit SVGD convergence bound and rate by choosing the step size sum to balance the terms on the right-hand side of (3). In particular, Corollary 2 instantiates an explicit SVGD step size sequence that drives the kernel Stein discrepancy to zero at an order $1/\sqrt{\log \log(n)}$ rate.

**Corollary 2** (A finite-particle convergence rate for SVGD)**.** *Instantiate the notation and assumptions of Theorem 3, let $(\bar{w}_{0,n}, \bar{A}, \bar{B}, \bar{C})$ be any upper bounds on $(w_{0,n}, A, B, C)$ respectively, and define the growth functions*

$$\phi(w) \triangleq \log \log(e^e + \tfrac{1}{w}) \quad \text{and} \quad \psi_{\bar{B}, \bar{C}}(x, y, \beta) \triangleq \tfrac{1}{\bar{C}} \log(\tfrac{1}{\bar{B}} \max(\bar{B}, \tfrac{1}{\beta} \log \tfrac{1}{x} - y)).$$

*If the step size sum $b_{t-1} = \sum_{r=0}^{t-1} \epsilon_r = s_n^\star$ for*

$$s_n^\star \triangleq \min \left( \psi_{\bar{B}, \bar{C}} \big( \bar{w}_{0,n} \sqrt{\phi(\bar{w}_{0,n})}, \bar{A}, \beta_1 \big), \psi_{\bar{B}, \bar{C}} \big( \bar{w}_{0,n} \phi(\bar{w}_{0,n}), \bar{A} + 2\bar{C}, \beta_2 \big) \right),$$

$$\beta_1 \triangleq \max \left( 1, \psi_{\bar{B}, \bar{C}} \big( \bar{w}_{0,n} \sqrt{\phi(\bar{w}_{0,n})}, \bar{A}, 1 \big) \right), \quad \text{and}$$

$$\beta_2 \triangleq \max \left( 1, \psi_{\bar{B}, \bar{C}} \big( \bar{w}_{0,n} \phi(\bar{w}_{0,n}), \bar{A} + 2\bar{C}, 1 \big) \right)$$

*then*

$$\min_{0 \leq r \leq t} \text{KSD}_P(\mu_r^n, P)$$

$$\leq \begin{cases} (\kappa_1 L + \kappa_2 d) \bar{w}_{0,n} + \kappa_1 d^{1/4} L \sqrt{2 M_{\mu_0^\infty, P} \bar{w}_{0,n}} + \sqrt{\frac{2}{R_{\alpha,1}} \text{KL}(\mu_0^\infty \| P)} & \text{if } s_n^\star = 0 \\ \frac{(\kappa_1 L + \kappa_2 d) + \kappa d^{1/4} L \sqrt{2 M_{\mu_0^\infty, P}}}{\sqrt{\phi(\bar{w}_{0,n})}} + \sqrt{\frac{2 \text{KL}(\mu_0^\infty \| P)}{R_{\alpha,1} + \frac{1}{\bar{C}} \log(\frac{1}{\bar{B}}(\frac{\log(1/(\bar{w}_{0,n} \phi(\bar{w}_{0,n})))}{\max(1, \psi_{\bar{B}, \bar{C}}(\bar{w}_{0,n}, 0, 1))} - \bar{A} - 2\bar{C}))}} & \text{otherwise} \end{cases} \quad (7)$$

$$= O \left( \frac{1}{\sqrt{\log \log(e^e + \frac{1}{\bar{w}_{0,n}})}} \right). \quad (8)$$

*If, in addition, $\mu_0^n = \frac{1}{n} \sum_{i=1}^n \delta_{x_i}$ for $x_i \overset{i.i.d.}{\sim} \mu_0^\infty$ with $M_{\mu_0^\infty} \triangleq \mathbb{E}_{\mu_0^\infty}[\|\cdot\|_2^2] < \infty$, then*

$$\bar{w}_{0,n} \triangleq \frac{M_{\mu_0^\infty} \log(n)^{\mathbb{I}[d=2]}}{\delta \, n^{1/(2 \vee d)}} \geq w_{0,n} \quad (9)$$

*with probability at least $1 - c\delta$ for a universal constant $c > 0$. Hence, with this choice of $\bar{w}_{0,n}$,*

$$\min_{0 \leq r \leq t} \text{KSD}_P(\mu_r^n, P) = O \left( \frac{1}{\sqrt{\log \log(n\delta)}} \right)$$

*with probability at least $1 - c\delta$.*

Specifically, given any upper bounds $(\bar{w}_{0,n}, \bar{A}, \bar{B}, \bar{C})$ on the quantities $(w_{0,n}, A, B, C)$ defined in Theorem 3, Corollary 2 specifies a recommended step size sum $s_n^\star$ to achieve an order $1/\sqrt{\log \log(e^e + \frac{1}{\bar{w}_{0,n}})}$ rate of SVGD convergence in KSD. Several remarks are in order. First,

the target step size sum $s_n^\star$ is easily computed given $(\bar{w}_{0,n}, \bar{A}, \bar{B}, \bar{C})$. Second, we note that the target $s_n^\star$ can equal $0$ when the initial Wasserstein upper bound $\bar{w}_{0,n}$ is insufficiently small since $\log(\frac{1}{\bar{B}}\max(\bar{B}, \frac{1}{\beta}\log\frac{1}{x}-y)) = \log(\frac{\bar{B}}{\bar{B}}) = 0$ for small arguments $x$. In this case, the setting $b_{t-1} = 0$ amounts to not running SVGD at all or, equivalently, to setting all step sizes to $0$.

Third, Corollary 2 also implies a time complexity for achieving an order $1/\sqrt{\log\log(e^e + \frac{1}{\bar{w}_{0,n}})}$ error rate. Recall that Theorem 3 assumes that $\max_{0 \leq r < t} \epsilon_r \leq R_{\alpha,1}$ for $R_{\alpha,1}$ defined in (1). Hence, $t^\star \triangleq \lceil s_n^\star/R_{\alpha,1}\rceil$ rounds of SVGD are necessary to achieve the recommended setting $\sum_{r=0}^{t^\star-1}\epsilon_r = s_n^\star$ while also satisfying the constraints of Theorem 3. Moreover, $t^\star$ rounds are also sufficient if each step size is chosen equal to $s_n^\star/t^\star$. In addition, since $s_n^\star = O(\log\log(e^e + \frac{1}{\bar{w}_{0,n}}))$, Corollary 2 implies that SVGD can deliver $\min_{0 \leq r \leq t^\star} \mathrm{KSD}_P(\mu_r^n, P) \leq \Delta$ in $t^\star = O(1/\Delta^2)$ rounds. Since the computational complexity of each SVGD round is dominated by $\Theta(n^2)$ kernel gradient evaluations (i.e., evaluating $\nabla_x k(x_i, x_j)$ for each pair of particles $(x_i, x_j)$), the overall computational complexity to achieve the order $1/\sqrt{\log\log(e^e + \frac{1}{\bar{w}_{0,n}})}$ error rate is $O(n^2\lceil s_n^\star/R_{\alpha,1}\rceil) = O(n^2\log\log(e^e + \frac{1}{\bar{w}_{0,n}}))$.

## 6 Proof of Theorem 1: Wasserstein discretization error of SVGD

In order to leverage Lemma 2, we first show that the pseudo-Lipschitzness conditions of Lemma 2 hold given our Assumptions 1 to 3. Recall that $s_p$ is Lipschitz and $x^\star$ satisfies $s_p(x^\star) = 0$ by Assumption 1. Then, by the triangle inequality, the definition of $\|\cdot\|_{\mathrm{op}}$, $\|k(x,z)\|_2 \leq \kappa_1^2$ and $\|\nabla_z\nabla_x k(x,z)\|_{\mathrm{op}} \leq \kappa_1^2$ from Assumption 2, and Cauchy-Schwartz,

$$\mathrm{Lip}(s_p(x)k(x,\cdot) + \nabla_x k(x,\cdot))$$
$$\leq \|s_p(x) - s_p(x^\star)\|_2\|\nabla_z k(x,z)\|_2 + \|\nabla_z\nabla_x k(x,z)\|_{\mathrm{op}}$$
$$= \sup_{\|u\|_2 \leq 1}(\|s_p(x) - s_p(x^\star)\|_2|\nabla_z k(x,z)^\top u|) + \|\nabla_z\nabla_x k(x,z)\|_{\mathrm{op}}$$
$$\leq L\|x - x^\star\|_2\|\nabla_z k(x,z)\|_2 + \|\nabla_z\nabla_x k(x,z)\|_{\mathrm{op}}$$
$$\leq \max(\kappa_1^2 L, \kappa_1^2)(1 + \|x - x^\star\|_2).$$

Letting $c_1 = \max(\kappa_1^2 L, \kappa_1^2)$ and taking supremum over $z$ proves the first pseudo-Lipschitzness condition. Similarly, we have

$$\mathrm{Lip}(s_p k(\cdot, z) + \nabla_x k(\cdot, z))$$
$$\leq \sup_{x\in\mathbb{R}^d}\mathrm{Lip}(s_p)|k(x,z)| + \|s_p(x) - s_p(x^\star)\|_2\|\nabla_x k(x,z)\|_2 + \|\nabla_x^2 k(x,z)\|_{\mathrm{op}}$$
$$\leq \kappa_1^2 L + \sup_{x\in\mathbb{R}^d}L\|x - x^\star\|_2\|\nabla_x k(x,z)\|_2 + \kappa_1^2, \tag{10}$$

where we used the Lipschitzness of $s_p$ from Assumption 1 and $|k(x,z)| \leq \kappa_1^2$, $\|\nabla_x^2 k(x,z)\|_{\mathrm{op}} \leq \kappa_1^2$ from Assumption 2. Now we consider two cases separately: when $\|x - z\|_2 \geq 1$ and $\|x - z\|_2 < 1$.

- Case 1: $\|x - z\|_2 \geq 1$. Recall that there exists $\gamma > 0$ such that $\|\nabla_x k(x,z)\|_2 \leq \gamma/\|x - z\|_2$ by Assumption 3. Then, using this together with the triangle inequality, we have

$$\|x - x^\star\|_2\|\nabla_x k(x,z)\|_2 \leq \gamma\frac{\|x-z\|_2 + \|z-x^\star\|_2}{\|x-z\|_2} \leq \gamma(1 + \|z - x^\star\|_2). \tag{11}$$

- Case 2: $\|x - z\|_2 < 1$. Then, using $\|\nabla_x k(x,z)\|_2 \leq \kappa_1^2$ from Assumption 2 and by the triangle inequality, we have

$$\|x - x^\star\|_2\|\nabla_x k(x,z)\|_2 \leq \kappa_1^2(\|x - z\|_2 + \|z - x^\star\|_2) < \kappa_1^2(1 + \|z - x^\star\|_2). \tag{12}$$

Combining (11) and (12) and using the triangle inequality, we get

$$\|x - x^\star\|_2\|\nabla_x k(x,z)\|_2 \leq \max(\gamma, \kappa_1^2)(1 + \|z - x^\star\|_2). \tag{13}$$

Plugging (13) back into (10), we can show the second pseudo-Lipschitzness condition holds for $c_2 = \max(\kappa_1^2(L + 1) + L\max(\gamma, \kappa_1^2), L\max(\gamma, \kappa_1^2)) = \kappa_1^2(L + 1) + L\max(\gamma, \kappa_1^2)$.

Now we have proved that both pseudo-Lipschitzness preconditions of Lemma 2 hold under our Assumptions 1 to 3. By repeated application of Lemma 2 and the inequality $(1+x) \leq e^x$, we have

$$W_1(\mu_{r+1}^n, \mu_{r+1}^\infty) = W_1(\Phi_{\epsilon_r}(\mu_r^n), \Phi_{\epsilon_r}(\mu_r^\infty)) \leq (1 + \epsilon_r D_r)W(\mu_r^n, \mu_r^\infty)$$

$$\leq W_1(\mu_0^n, \mu_0^\infty)\prod_{s=0}^r (1 + \epsilon_s D_s) \leq W_1(\mu_0^n, \mu_0^\infty)\exp\left(\sum_{s=0}^r \epsilon_s D_s\right) \quad (14)$$

for $D_s = c_1(1 + m_{\mu_s^n, x^\star}) + c_2(1 + m_{\mu_s^\infty, x^\star})$.

Using the result from Lemma 3, we have

$$D_{s+1} \leq A + B\exp(Cb_s)$$

for $A = (c_1 + c_2)(1 + m_{P,x^\star})$, $B = c_1 m_{\mu_0^n, P} + c_2 m_{\mu_0^\infty, P}$, and $C = \kappa_1^2(3L + d)$. Therefore

$$\sum_{s=0}^r \epsilon_s D_s \leq \max_{0 \leq s \leq r} D_s \sum_{s=0}^r \epsilon_s \leq b_r(A + B\exp(Cb_{r-1})) \leq b_r(A + B\exp(Cb_r)).$$

Plugging this back into (14) proves the result.

## 7 Proof of Lemma 3: SVGD moment growth

From Assumption 2 we know $|k(y,x)| \leq \kappa_1^2$ and $\left\|\nabla_y^2 k(y,x)\right\|_{op} \leq \kappa_1^2$. The latter implies

$$\left\|\nabla_y k(y,x) - \nabla_z k(z,x)\right\|_2 \leq \kappa_1^2 \|y - z\|_2.$$

Recall that $s_p$ is Lipschitz and satisfies $\mathbb{E}_P[s_p(\cdot)] = 0$ by Assumption 1. Let $\mu$ be any probability measure. Using the above results, Jensen's inequality and the fact that $\mathbb{E}_{Z \sim P}[(\mathcal{A}_P k(\cdot, x))(Z)] = 0$, we have

$$\|T_{\mu,\epsilon}(x) - x\|_2 \leq \epsilon\|\mathbb{E}_{X \sim \mu}[(\mathcal{A}_P k(\cdot, x))(X)]\|_2$$
$$= \epsilon\|\mathbb{E}_{X \sim \mu}[(\mathcal{A}_P k(\cdot, x))(X)] - \mathbb{E}_{Z \sim P}[(\mathcal{A}_P k(\cdot, x))(Z)]\|_2$$
$$= \epsilon\|\mathbb{E}_{(X,Z) \sim \mu \otimes P}[k(Z,x)(s_p(X) - s_p(Z)) + (k(X,x) - k(Z,x))(s_p(X) - \mathbb{E}_P[s_p(\cdot)])$$
$$\quad + (\nabla_X k(X,x) - \nabla_Z k(Z,x))]\|_2$$
$$\leq \epsilon\mathbb{E}_{(X,Z) \sim \mu \otimes P}[|k(Z,x)|\|s_p(X) - s_p(Z)\|_2 + (|k(X,x)| + |k(Z,x)|)\|s_p(X) - \mathbb{E}_{Y \sim P}[s_p(Y)]\|_2$$
$$\quad + \|\nabla_X k(X,x) - \nabla_Z k(Z,x)\|_2]$$
$$\leq \epsilon\mathbb{E}_{(X,Z) \sim \mu \otimes P}[\kappa_1^2(L+1)\|X - Z\|_2] + \epsilon \cdot 2\kappa_1^2 L\mathbb{E}_{(X,Y) \sim \mu \otimes P}[\|X - Y\|_2]$$
$$= \epsilon\kappa_1^2(3L+1)\mathbb{E}_{(X,Z) \sim \mu \otimes P}[\|X - Z\|_2]$$
$$= \epsilon C m_{\mu,P}. \quad (15)$$

The last step used the definitions $m_{\mu,P} \triangleq \mathbb{E}_{(X,Z) \sim \mu \otimes P}[\|X - Z\|_2]$ and $C = \kappa_1^2(3L+1)$. Then, applying the triangle inequality and (15), we have

$$m_{\mu_{r+1}, P} = \mathbb{E}_{(X,Z) \sim \mu_{r+1} \otimes P}[\|X - Z\|_2] = \mathbb{E}_{(X,Z) \sim \mu_r \otimes P}[\|T_{\mu_r, \epsilon_r}(X) - Z\|_2]$$
$$\leq \mathbb{E}_{(X,Z) \sim \mu_r \otimes P}[\|T_{\mu_r, \epsilon_r}(X) - X\|_2 + \|X - Z\|_2] \leq (1 + \epsilon_r C)m_{\mu_r, P}, \quad (16)$$
$$M_{\mu_{r+1}, P} = \mathbb{E}_{(X,Z) \sim \mu_{r+1} \otimes P}[\|X - Z\|_2^2] = \mathbb{E}_{(X,Z) \sim \mu_r \otimes P}[\|T_{\mu_r, \epsilon_r}(X) - Z\|_2^2]$$
$$\leq \mathbb{E}_{(X,Z) \sim \mu_r \otimes P}[\|T_{\mu_r, \epsilon_r}(X) - X\|_2^2 + 2\|T_{\mu_r, \epsilon_r}(X) - X\|_2\|X - Z\|_2 + \|X - Z\|_2^2]$$
$$\leq (\epsilon_r^2 C^2 + 2\epsilon_r C)m_{\mu_r, P}^2 + M_{\mu_r, P} \leq (1 + 2\epsilon_r C + \epsilon_r^2 C^2)M_{\mu_r, P}$$
$$= (1 + \epsilon_r C)^2 M_{\mu_r, P}, \quad (17)$$

where the second last step used Jensen's inequality $m_{\mu_r, P}^2 \leq M_{\mu_r, P}$. Then, we repeatedly apply (16) and (17) together with the triangle inequality and the bound $1 + x \leq e^x$ to get

$$M_{\mu_r, P} \leq M_{\mu_0, P}\prod_{s=0}^{r-1}(1 + \epsilon_s C)^2 \leq M_{\mu_0, P}\exp\left(2C\sum_{s=0}^{r-1} \epsilon_s\right) \leq M_{\mu_0, P}\exp(2Cb_{r-1}) \text{ and}$$

$$m_{\mu_r, x^\star} - m_{P, x^\star} \leq m_{\mu_r, P} \leq m_{\mu_0, P}\prod_{s=0}^{r-1}(1 + \epsilon_s C) \leq m_{\mu_0, P}\exp(Cb_{r-1}).$$

# 8 Proof of Lemma 4: KSD-Wasserstein bound

Our proof generalizes that of Gorham and Mackey [9, Lem. 18]. Consider any $g \in \mathcal{H}^d$ satisfying $\|g\|_{\mathcal{H}^d}^2 \triangleq \sum_{i=1}^d \|g_i\|_{\mathcal{H}}^2 \le 1$. From Assumption 2 we know

$$\|g(x)\|_2^2 \le k(x,x) \sum_{i=1}^d \|g_i\|_{\mathcal{H}}^2 \le \kappa_1^2, \tag{18}$$

$$\|\nabla g(x)\|_{op}^2 \le \|\nabla g(x)\|_F^2 = \sum_{i=1}^d \sum_{j=1}^d |\nabla_{x_i} g_j(x)|^2 \le \|g\|_{\mathcal{H}^d}^2 \operatorname{tr}(\nabla_y \nabla_x k(x,y)|_{y=x})$$

$$\le d\|\nabla_y \nabla_x k(x,y)|_{y=x}\|_{op} \le \kappa_1^2 d, \text{ and} \tag{19}$$

$$\|\nabla(\nabla \cdot g(x))\|_2^2 = \sum_{i=1}^d \left(\sum_{j=1}^d \nabla_{x_i} \nabla_{x_j} g_j(x)\right)^2 \le d \sum_{i=1}^d \sum_{j=1}^d |\nabla_{x_i} \nabla_{x_j} g_j(x)|^2$$

$$\le d \sum_{i=1}^d \sum_{j=1}^d \|g_j\|_{\mathcal{H}}^2 (\nabla_{y_i} \nabla_{y_j} \nabla_{x_i} \nabla_{x_j} k(x,y)|_{y=x}) \le \kappa_2^2 d^2,$$

Suppose $X, Y, Z$ are distributed so that $(X, Y)$ is a 1-Wasserstein optimal coupling of $(\mu, \nu)$ and $Z$ is independent of $(X, Y)$. Since $s_p$ is $L$-Lipschitz with $\mathbb{E}_P[s_p] = 0$ (Assumption 1), $g$ is bounded (18), and $g$ and $\nabla \cdot g$ are Lipschitz (19), repeated use of Cauchy-Schwarz gives

$$\mathbb{E}_\mu[\mathcal{T}_P g] - \mathbb{E}_\nu[\mathcal{T}_P g]$$
$$= \mathbb{E}[\nabla \cdot g(X) - \nabla \cdot g(Y)] + \mathbb{E}[\langle s_p(X) - s_p(Y), g(X)\rangle] + \mathbb{E}[\langle s_p(Y) - s_p(Z), g(X) - g(Y)\rangle]$$
$$\le (\kappa_2 d + \kappa_1 L) W_1(\mu, \nu) + L\mathbb{E}[\|Y - Z\|_2 \min(2\kappa_1, \kappa_1\sqrt{d}\|X - Y\|_2)].$$

Since our choice of $g$ was arbitrary, the first advertised result now follows from the definition of KSD (Definition 4). The second claim then follows from Cauchy-Schwarz and the inequality $\min(a,b)^2 \le ab$ for $a, b \ge 0$, since

$$\mathbb{E}[\|Y - Z\|_2 \min(2\kappa_1, \kappa_1\sqrt{d}\|X - Y\|_2)] \le M_{\nu,P}^{1/2} \mathbb{E}[\min(2\kappa_1, \kappa_1\sqrt{d}\|X - Y\|_2)^2]^{1/2}$$
$$\le \sqrt{2M_{\nu,P}} \kappa_1 d^{1/4} \mathbb{E}[\|X - Y\|_2]^{1/2} = \sqrt{2M_{\nu,P} W_1(\mu, \nu)} \kappa_1 d^{1/4}.$$

# 9 Conclusions and Limitations

In summary, we have proved the first unified convergence bound and rate for finite-particle SVGD. In particular, our results show that with a suitably chosen step size sequence, SVGD with $n$-particles drives the KSD to zero at an order $1/\sqrt{\log\log(n)}$ rate. The assumptions we have made on the target and kernel are mild and strictly weaker than those used in prior work to establish KSD weak convergence control [9, 3, 12, 1]. However, we suspect that, with additional effort, the Lipschitz score assumption (Assumption 1) can be relaxed to accommodate pseudo-Lipschitz scores as in Erdogdu et al. [7] or weakly-smooth scores as in Sun et al. [24]. A second limitation of this work is that the obtained rate of convergence is quite slow. However, we hope that this initial recipe for explicit, non-asymptotic convergence will serve as both a template and a catalyst for the field to develop refined upper and lower bounds for SVGD error. To this end, we leave the reader with several open challenges. First, can one establish a non-trivial minimax lower bound for the convergence of SVGD? Second, can one identify which types of target distributions lead to worst-case convergence behavior for SVGD? Finally, can one identify commonly met assumptions on the target distribution and kernel under which the guaranteed convergence rate of SVGD can be significantly improved? Promising follow-up work has already begun investigating speed-ups obtainable by focusing on the convergence of a finite set of moments [20] or by modifying the SVGD algorithm [5].

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

# A  Kernel Assumptions

To show that Assumptions 2 and 3 are met by the most commonly used SVGD kernels with constants independent of dimension, we begin by bounding the derivatives of any radial kernel of the form $k(x, y) = \phi(\|x - y\|_2^2/2)$ with $\phi : \mathbb{R} \to \mathbb{R}$ four times differentiable. By the reproducing property and Cauchy-Schwarz we have

$$|k(x,y)| = |\langle k(x,\cdot), k(y,\cdot)\rangle_{\mathcal{H}}| \leq \|k(x,\cdot)\|_{\mathcal{H}}\|k(y,\cdot)\|_{\mathcal{H}} = \sqrt{k(x,x)}\sqrt{k(y,y)} = \phi(0),$$

$$\|\nabla_x k(x,y)\|_2 = |\phi'(\|x-y\|_2^2/2)|\|x-y\|_2,$$

$$\|\nabla_y \nabla_x k(x,y)\|_{\mathrm{op}} = \left\|-\phi''(\|x-y\|_2^2/2)(x-y)(x-y)^\top - \phi'(\|x-y\|_2^2/2)I\right\|_{\mathrm{op}}$$

$$\leq |\phi''(\|x-y\|_2^2/2)|\|x-y\|_2^2 + |\phi'(\|x-y\|_2^2/2)|, \quad \text{and}$$

$$\left\|\nabla_x^2 k(x,y)\right\|_{\mathrm{op}} = \left\|\phi''(\|x-y\|_2^2/2)(x-y)(x-y)^\top + \phi'(\|x-y\|_2^2/2)I\right\|_{\mathrm{op}}$$

$$\leq |\phi''(\|x-y\|_2^2/2)|\|x-y\|_2^2 + |\phi'(\|x-y\|_2^2/2)|.$$

Similarly, the partial derivatives take the form

$$\nabla_{x_j} k(x,y) = \phi'(\|x-y\|_2^2/2)(x_j - y_j)$$

$$\nabla_{y_j}\nabla_{x_j} k(x,y) = -\phi'(\|x-y\|_2^2/2) - \phi''(\|x-y\|_2^2/2)(y_j - x_j)^2$$

$$\nabla_{x_i}\nabla_{y_j}\nabla_{x_j} k(x,y) = -\phi''(\|x-y\|_2^2/2)(x_i - y_i) - \phi'''(\|x-y\|_2^2/2)(y_j - x_j)^2(x_i - y_i)$$
$$- \mathbb{I}[i = j]2\phi''(\|x-y\|_2^2/2)(x_i - y_i)$$
$$= -\phi''(\|x-y\|_2^2/2)(x_i - y_i)$$
$$- \mathbb{I}[i \neq j]\phi'''(\|x-y\|_2^2/2)(y_j - x_j)^2(x_i - y_i)$$
$$+ \mathbb{I}[i = j](\phi'''(\|x-y\|_2^2/2)(y_i - x_i)^3 - 2\phi''(\|x-y\|_2^2/2)(x_i - y_i))$$

$$\nabla_{y_i}\nabla_{x_i}\nabla_{y_j}\nabla_{x_j} k(x,y) = \phi'''(\|x-y\|_2^2/2)(x_i - y_i)^2 + \phi''(\|x-y\|_2^2/2)$$
$$+ \mathbb{I}[i \neq j](\phi''''(\|x-y\|_2^2/2)(y_j - x_j)^2(x_i - y_i)^2$$
$$+ \phi'''(\|x-y\|_2^2/2)(y_j - x_j)^2)$$
$$+ \mathbb{I}[i = j](\phi''''(\|x-y\|_2^2/2)(y_i - x_i)^4 + 5\phi'''(\|x-y\|_2^2/2)(y_i - x_i)^2$$
$$+ 2\phi''(\|x-y\|_2^2/2))$$
$$= \phi'''(\|x-y\|_2^2/2)((x_i - y_i)^2 + (x_j - y_j)^2) + \phi''(\|x-y\|_2^2/2)$$
$$+ \phi''''(\|x-y\|_2^2/2)(y_j - x_j)^2(x_i - y_i)^2$$
$$+ \mathbb{I}[i = j](4\phi'''(\|x-y\|_2^2/2)(y_i - x_i)^2 + 2\phi''(\|x-y\|_2^2/2))$$

so that both $|k(x,y)|$ and

$$\nabla_{y_i}\nabla_{x_i}\nabla_{y_j}\nabla_{x_j} k(x,y)|_{y=x} = \phi''(0) + \mathbb{I}[i = j]2\phi''(0)$$

are bounded (Assumption 2) by constants independent of dimension.

Gorham and Mackey [9] popularized the use of IMQ kernels for SVGD, by establishing the convergence-determining properties of the associated KSD. The corresponding $\phi$ satisfies

$$\phi(t) = (c^2 + 2t)^\beta \quad \text{for} \quad c > 0 \quad \text{and} \quad \beta \in (-1, 0),$$
$$\phi'(t) = 2\beta(c^2 + 2t)^{\beta-1}, \quad \text{and} \quad \phi''(t) = 4\beta(\beta - 1)(c^2 + 2t)^{\beta-2}.$$

In this case, $\|\nabla_y \nabla_x k(x,y)\|_{\mathrm{op}}$ and $\left\|\nabla_x^2 k(x,y)\right\|_{\mathrm{op}}$ are bounded (Assumption 2) by constants independent of dimension as

$$|\phi'(\|x-y\|_2^2/2)| = -2\beta(c^2 + \|x-y\|_2^2)^{\beta-1}$$
$$\leq -2\beta\min(c^{2\beta-2}, \|x-y\|_2^{2\beta-2}) \leq -2\beta c^{2\beta-2} \quad \text{and}$$

$$|\phi''(\|x-y\|_2/2)|\|x-y\|_2^2 = 4\beta(\beta - 1)(c^2 + \|x-y\|_2^2)^{\beta-2}\|x-y\|_2^2$$
$$\leq 4\beta(\beta - 1)(c^2 + \|x-y\|_2^2)^{\beta-1} \leq 4\beta(\beta - 1)c^{2\beta-2}.$$

For $\|\nabla_x k(x,y)\|_2$, we consider two cases:

- When $\|x - y\|_2 \geq 1$,

$$|\phi'(\|x - y\|_2^2/2)|\|x - y\|_2 \leq -2\beta\|x - y\|_2^{2\beta-2}\|x - y\|_2 = -2\beta\|x - y\|_2^{2\beta-1}$$
$$\leq -2\beta/\|x - y\|_2 \leq -2\beta.$$

- When $\|x - y\|_2 < 1$, $|\phi'(\|x - y\|_2^2/2)|\|x - y\|_2 < |\phi'(\|x - y\|_2^2/2)| \leq -2\beta c^{2\beta-2}$.

Therefore, $\|\nabla_x k(x, y)\|_2$ is also bounded (Assumption 2) by constants independent of dimension, and Assumption 3 holds with $\gamma = -2\beta$.

The original SVGD paper [18] used Gaussian kernels in all experiments, and they remain perhaps the most common choice in the literature. In this case, $\phi$ satisfies

$$\phi(t) = e^{-2\alpha t} \quad \text{for} \quad \alpha > 0, \quad \phi'(t) = -2\alpha e^{-2\alpha t} = -2\alpha\phi(t), \quad \text{and} \quad \phi''(t) = 4\alpha^2\phi(t).$$

Using the inequality $x \leq e^{x-1}$, we find that

$$|\phi'(\|x - y\|_2^2/2)| = 2\alpha e^{-\alpha\|x-y\|_2^2} \leq \min(2\alpha, 2/(e\|x - y\|_2^2)) \quad \text{and}$$
$$|\phi''(\|x - y\|_2/2)|\|x - y\|_2^2 = 4\alpha^2 e^{-\alpha\|x-y\|_2^2}\|x - y\|_2^2 \leq 4\alpha/e$$

so that $\|\nabla_y \nabla_x k(x, y)\|_{\mathrm{op}}$, $\|\nabla_x^2 k(x, y)\|_{\mathrm{op}}$, and $\|\nabla_x k(x, y)\|_2$ are bounded (Assumption 2) by constants independent of dimension, and Assumption 3 holds with $\gamma = 2/e$.

# B  Proof of Corollary 2: A finite-particle convergence rate for SVGD

We begin by establishing a lower bound on $b_{t-1}$. Let

$$b_{t-1}^{(1)} = \psi_{\bar{B},\bar{C}}(\bar{w}_{0,n}\sqrt{\phi(\bar{w}_{0,n})}, \bar{A}, \beta_1) \quad \text{and} \quad b_{t-1}^{(2)} = \psi_{\bar{B},\bar{C}}(\bar{w}_{0,n}\,\phi(\bar{w}_{0,n}), \bar{A} + 2\bar{C}, \beta_2)$$

so that $b_{t-1} = \min(b_{t-1}^{(1)}, b_{t-1}^{(2)})$. Since $\beta_1, \beta_2, \phi(\bar{w}_{0,n}) \geq 1$, we have

$$\beta_1 = \max(1, \tfrac{1}{\bar{C}}\log(\tfrac{1}{\bar{B}}(\log \tfrac{1}{\bar{w}_{0,n}\sqrt{\phi(\bar{w}_{0,n})}} - \bar{A})))$$
$$\leq \max(1, \tfrac{1}{\bar{C}}\log(\tfrac{1}{\bar{B}}(\log \tfrac{1}{\bar{w}_{0,n}\sqrt{\phi(\bar{w}_{0,n})}})))$$
$$\leq \max(1, \tfrac{1}{\bar{C}}\log(\tfrac{1}{\bar{B}}(\log \tfrac{1}{\bar{w}_{0,n}}))) \quad \text{and}$$
$$\beta_2 = \max(1, \tfrac{1}{\bar{C}}\log(\tfrac{1}{\bar{B}}(\log \tfrac{1}{\bar{w}_{0,n}\phi(\bar{w}_{0,n})} - \bar{A} - 2\bar{C})))$$
$$\leq \max(1, \tfrac{1}{\bar{C}}\log(\tfrac{1}{\bar{B}}(\log \tfrac{1}{\bar{w}_{0,n}\phi(\bar{w}_{0,n})})))$$
$$\leq \max(1, \tfrac{1}{\bar{C}}\log(\tfrac{1}{\bar{B}}(\log \tfrac{1}{\bar{w}_{0,n}}))).$$

Hence, $\phi(\bar{w}_{0,n}) \geq 1$ implies that

$$b_{t-1}^{(1)} \geq \tfrac{1}{\bar{C}}\log(\tfrac{1}{\bar{B}}(\frac{\log \frac{1}{\bar{w}_{0,n}\sqrt{\phi(\bar{w}_{0,n})}}}{\max(1, \tfrac{1}{\bar{C}}\log(\tfrac{1}{\bar{B}}(\log \tfrac{1}{\bar{w}_{0,n}})))} - \bar{A}))$$

$$\geq \tfrac{1}{\bar{C}}\log(\tfrac{1}{\bar{B}}(\frac{\log \frac{1}{\bar{w}_{0,n}\,\phi(\bar{w}_{0,n})}}{\max(1, \tfrac{1}{\bar{C}}\log(\tfrac{1}{\bar{B}}(\log \tfrac{1}{\bar{w}_{0,n}})))} - \bar{A} - 2\bar{C})) \quad \text{and} \tag{20}$$

$$b_{t-1}^{(2)} \geq \tfrac{1}{\bar{C}}\log(\tfrac{1}{\bar{B}}(\frac{\log \frac{1}{\bar{w}_{0,n}\,\phi(\bar{w}_{0,n})}}{\max(1, \tfrac{1}{\bar{C}}\log(\tfrac{1}{\bar{B}}(\log \tfrac{1}{\bar{w}_{0,n}})))} - \bar{A} - 2\bar{C})).$$

We divide the remainder of our proof into four parts. First we prove each of the two cases in the generic KSD bound (7) in Appendices B.1 and B.2. Next we show in Appendix B.3 that these two cases yield the generic convergence rate (8). Finally, we prove the high probability upper estimate (9) for $w_{0,n}$ under i.i.d. initialization in Appendix B.4.

## B.1  Case $b_{t-1} = 0$

In this case, the error bound (7) follows directly from Theorem 3.

## B.2 Case $b_{t-1} > 0$

We first state and prove a useful lemma.

**Lemma 6.** *Suppose $x = f(\beta)$ for a non-increasing function $f : \mathbb{R} \to \mathbb{R}$ and $\beta = \max(1, f(1))$. Then $x \leq \beta$ and $x \leq f(x)$.*

*Proof.* Because $f$ is non-increasing and $\beta \geq 1$, $x = f(\beta) \leq f(1) \leq \beta$ . Since $x \leq \beta$ and $f$ is non-increasing, we further have $f(x) \geq f(\beta) = x$ as advertised. $\square$

Since $\psi_{\bar{B}, \bar{C}}$ is non-increasing in its third argument, Lemma 6 implies that $b_{t-1}^{(1)} \leq \beta_1$ and

$$b_{t-1}^{(1)} \leq \psi_{\bar{B}, \bar{C}}(\bar{w}_{0,n}\sqrt{\phi(\bar{w}_{0,n})}, \bar{A}, b_{t-1}^{(1)}).$$

Rearranging the terms and noting that

$$\bar{B} < \tfrac{1}{\beta_1} \log \tfrac{1}{\bar{w}_{0,n}\sqrt{\phi(\bar{w}_{0,n})}} - \bar{A} \leq \tfrac{1}{b_{t-1}^{(1)}} \log \tfrac{1}{\bar{w}_{0,n}\sqrt{\phi(\bar{w}_{0,n})}} - \bar{A}$$

since $b_{t-1}^{(1)} \geq b_{t-1} > 0$, we have

$$\bar{w}_{0,n} \exp(b_{t-1}^{(1)}(\bar{A} + \bar{B}\exp(\bar{C}b_{t-1}^{(1)}))) \leq \tfrac{1}{\sqrt{\phi(\bar{w}_{0,n})}}. \tag{21}$$

Similarly, we have $b_{t-1}^{(2)} \leq \psi_{\bar{B}, \bar{C}}(\bar{w}_{0,n} \log\log \tfrac{1}{\bar{w}_{0,n}}, \bar{A} + 2\bar{C}, b_{t-1}^{(2)})$ and

$$\sqrt{\bar{w}_{0,n}} \exp(b_{t-1}^{(2)}(2\bar{C} + \bar{A} + \bar{B}\exp(\bar{C}b_{t-1}^{(2)}))/2) \leq \tfrac{1}{\sqrt{\phi(\bar{w}_{0,n})}}. \tag{22}$$

Since $b_{t-1} = \min(b_{t-1}^{(1)}, b_{t-1}^{(2)})$, the inequalities (21) and (22) are also satisfied when $b_t$ is substituted for $b_{t-1}^{(1)}$ and $b_{t-1}^{(2)}$. Since the error term $a_{t-1}$ (4) is non-decreasing in each of $(w_{0,n}, A, B, C)$, we have

$$a_{t-1} \leq (\kappa_1 L + \kappa_2 d + \kappa_1 d^{1/4} L \sqrt{2M_{\mu_0^\infty, P}})/\sqrt{\phi(\bar{w}_{0,n})}.$$

Since $b_{t-1} = \min(b_{t-1}^{(1)}, b_{t-1}^{(2)})$, the claim (7) follows from this estimate, the lower bounds (20), and Theorem 3.

## B.3 Generic convergence rate

The generic convergence rate (8) holds as, by the lower bounds (20), $b_{t-1} = \min(b_{t-1}^{(1)}, b_{t-1}^{(2)}) > 0$ whenever

$$e^{-(\bar{B}+\bar{A}+2\bar{C})} > \bar{w}_{0,n}\phi(\bar{w}_{0,n}) \quad \text{and} \quad \bar{B}^{(\bar{B}+\bar{A}+2\bar{C})/\bar{C}} > \bar{w}_{0,n}\phi(\bar{w}_{0,n})(\log(1/\bar{w}_{0,n}))^{(\bar{B}+\bar{A}+2\bar{C})/\bar{C}},$$

a condition which occurs whenever $\bar{w}_{0,n}$ is sufficiently small since the right-hand side of each inequality converges to zero as $\bar{w}_{0,n} \to 0$.

## B.4 Initializing with i.i.d. particles

We begin by restating an expected Wasserstein bound due to Lei [16].

**Lemma 7** (Lei [16, Thm. 3.1]). *Suppose $\mu_0^n = \tfrac{1}{n}\sum_{i=1}^n \delta_{x_i}$ for $x_i \overset{i.i.d.}{\sim} \mu_0^\infty$ with $M_{\mu_0^\infty} \triangleq \mathbb{E}_{\mu_0^\infty}[\|\cdot\|_2^2] < \infty$. Then, for a universal constant $c > 0$,*

$$\mathbb{E}[W_1(\mu_0^n, \mu_0^\infty)] \leq cM_{\mu_0^\infty} \frac{\log(n)^{\mathbb{I}[d=2]}}{n^{1/(2\vee d)}}.$$

Together, Lemma 7 and Markov's inequality imply that

$$W_1(\mu_0^n, \mu_0^\infty) \leq \mathbb{E}[W_1(\mu_0^n, \mu_0^\infty)]/(c\delta) \leq M_{\mu_0^\infty} \tfrac{\log(n)^{\mathbb{I}[d=2]}}{n^{1/(2\vee d)}}/\delta$$

with probability at least $1 - c\delta$, proving the high probability upper estimate (9).

