# OpenReview forum: "A Finite-Particle Convergence Rate for Stein Variational Gradient Descent"
_NeurIPS.cc/2023/Conference — NeurIPS 2023 poster_

### Official Review · Reviewer_awG8 · 2023-07-03

**Soundness:** 3 good
**Presentation:** 4 excellent
**Contribution:** 3 good
**Rating:** 7
**Confidence:** 2

**Summary:**

This paper provides an analysis of the convergence rate of finite-sample Stein Variational Gradient Descent (SVGD) for sub-Gaussian targets with Lipschitz scores. In contrast to previous works such as Liu 2017, Duncan et al. 2019 and Korba et al. 2020, the presented results offer convergence guarantees that hold for finite samples and rely on weaker assumptions. The authors present two key results regarding the discretization error of finite-sample SVGD compared to infinite-sample SVGD, measured in terms of Wasserstein (Theorem 1) and KSD (Theorem 2), respectively. These results are then combined to establish a finite-sample bound on the KSD error between a finite-sample approximation and the target distribution (Theorem 3). By carefully selecting step sizes, the authors demonstrate that this error decays at a rate of $1 / \sqrt{\log \log n}$ (Corollary 2).

**Strengths:**

**Quality**: This paper provides a rigorous study of the convergence of finite-sample SVGD and delivers clear and well-supported results. The main findings (Theorem 1, 2, 3) build upon existing results (Lemma 1 and 4) but require sophisticated combinations and detailed arguments. The authors provide extensive discussions and overview the proof strategy, which appears reasonable and comprehensive (although I did not examine the proofs in depth).

**Novelty**: This paper offers a crucial contribution by bridging the gap between the existing convergence guarantees of infinite-sample SVGD and the practical implementation of finite-sample approximations, which is currently lacking in the literature. Whilst the established convergence rate is slow, this work stands out as the first to provide an explicit, non-asymptotic guarantee for SVGD approximations under reasonably mild assumptions on the target distribution. This, in my view, is the primary novelty and significance of this paper.

**Clarity**: The paper demonstrates clear motivation and objectives. Despite the technical nature of the content, the exposition is highly comprehensible. The intuitive explanations and remarks provided before and after each main theorem are particularly valuable in aiding understanding.

**Weaknesses:**

**Slow convergence rate**: As mentioned above and in the paper, the established rate $1 / \sqrt{\log \log n}$  is notably slow, suggesting that the bound (3) is likely to be very loose. Although the authors acknowledge the potential of the presented proof strategy as a starting point for refining the bounds, it is not entirely clear whether or not or how the proof could be adapted to achieve an improved bound. Including discussions on potential avenues for enhancing the convergence rate would be valuable and insightful.

**Questions:**

1. There are a few places that say “the Algorithm 2 outputs $\mu_r^n = SVGD(\mu_0^n, r)$…”, e.g. L166. Since this output is based on a discrete measure $\mu_0^n$, should it be “the **Algorithm 1** outputs …” instead?
2. COuld you elaborate on the suggested step size scheme in Corollary 2? Specifically, what is the dependence on the sample size $n$ and on the dimension $d$? How easy is it to construct the upper bounds $(\overline{w}_{0, n}, \overline{A}, \overline{B}, \overline{C})$ in practice?
3. As also mentioned in the paper, the established rate $1 / \sqrt{\log \log n}$ is very slow. Could you provide some insights into which parts of the proof strategy could possibly have led to this slow rate? Is the bound (3) tight? How does this convergence rate compare with empirical performance in numerical simulations?

**Limitations:**

The assumptions on the target distribution, its score function and the positive definite kernel are summarised in Section 2, accompanied by interpretations and discussions on their connections to related literature. Limitations are also discussed in Section 9. Overall, I think the discussions on assumptions and limitations are adequately covered.

---

> ### Author Rebuttal · Authors · 2023-08-10
>
> We thank the reviewer for the positive feedback. We are pleased that the reviewer found our proof strategy reasonable, our contribution crucial, and our exposition comprehensive. We respond to the detailed comments below.
>
> ### “the Algorithm 2 outputs …”, e.g. L166
> Thank you for pointing that out! Indeed, it should be Algorithm 1.
>
> ### “Could you elaborate on the suggested step size scheme in Corollary 2?”
>
> The step size scheme is constructed such that the two terms in the unified error bound (Theorem 3) balance with each other as there is a tradeoff between them—the discretization error bound (i.e., $a_{t-1}$) grows with the step size sum while the continuous SVGD error decreases proportional to it. The optimal step size sum is
>
> $O(\\log\\log(e^e+\\frac{1}{ \\bar{w}_{0,n} })),$
>
> and after plugging in the upper bound in eq. (9):
>
> $O(\\log\\log(e^e + \\frac{ \\delta n^{1/(2\\vee d)} }{ M_{\\mu_0^{\\infty}}\\log(n)^{\\mathbf{1}[d=2]} }) ).$
>
> As is used above, an upper bound for $w_{0,n}$ is given in eq. (9).
> To get upper bounds $\\bar{A}, \\bar{B}, \\bar{C}$ it suffices to produce the following upper bounds:
> * Kernel constant upper bounds: $\\kappa_1, \\kappa_2, \\gamma$. These are straightforward to compute explicitly, and we have provided these values in Appendix A for Gaussian and IMQ kernels.
> * Lipschitz constant of the score upper bound: This can be derived from the observable score function (which does not require knowledge of the normalizing constant of the target density) and is a standard input to score-based distributional approximation like Langevin Monte Carlo.
> * Moment upper bounds: It suffices to provide any upper bound on $\\mathbb{E}_{P}[\\|\\cdot\\|_2]$ as
>
> $m_{P, x^*} \\leq \\|x^*\\|_2 +  \\mathbb{E}_P [\\|\\cdot\\|_2],$
>
> $m_{\\mu_0, x^*} \\leq \\mathbb{E}_{\\mu_0}[\\|\\cdot\\|_2] +  \\mathbb{E}_P [\\|\\cdot\\|_2],$
>
> can $x^*$ be identified efficiently by running gradient ascent to find a stationary point, and $\\mathbb{E}_{\\mu_0}[\\|\\cdot\\|_2]$ can be numerically estimated.
>
> ### Is the bound (3) tight? How does this convergence rate compare with empirical performance in numerical simulations?
>
> We believe the first part of the bound ($a_{t-1}$) can be further improved (please see our response to questions shared by the reviewers for detailed reasons) and as suggested in the submission we expect our worst case rate (which holds for any initialization and a broad class of target distributions and step size sequences) to be slower than the true convergence rate observed in practice for many distributions, initializations, and step size settings. We believe that developing a non-trivial lower bound for SVGD performance to assess tightness is an important open question.

---

> > ### Comment · Reviewer_awG8 · 2023-08-12
> >
> > Thank you for your response, which have answered my questions. I would like to keep my scores.

---

### Official Review · Reviewer_bq3E · 2023-07-04

**Soundness:** 2 fair
**Presentation:** 2 fair
**Contribution:** 2 fair
**Rating:** 5
**Confidence:** 1

**Summary:**

In this work, the authors present a novel analysis of finite-particle Stein Variational Gradient Descent (SVGD) and derive a unified convergence bound for this algorithm. The convergence bound provides an explicit measure of how close the finite-particle SVGD algorithm gets to its target. To establish this convergence bound, the authors first introduce a bound on the discretization error of the 1-Wasserstein distance between the finite-particle and continuous SVGD. They make certain assumptions that are commonly satisfied in SVGD applications and compatible with Kernelized Stein Discrepancy (KSD) weak convergence control.

Overall, this work contributes to the understanding of finite-particle SVGD and provides a unified convergence bound that quantifies the algorithm's convergence to its target. The derived bounds enable better control and evaluation of the accuracy of finite-particle SVGD in practical applications.

**Strengths:**

The strengths are:
* The authors have proved the first unified convergence bound and rate for finite-particle SVGD.
* They show that SVGD with n-particles drives the KSD to zero at an order $1/ \sqrt{\log\log(n)}$ rate.

**Weaknesses:**

In my perspective, this paper would be more suitable for an optimization journal, as it would provide an environment where the technical contributions of the study can undergo a more comprehensive evaluation through an extended rebuttal cycle.

**Questions:**

None.

**Limitations:**

None.

---

> ### Author Rebuttal · Authors · 2023-08-10
>
> We thank the reviewer for acknowledging our contributions and for the feedback. We respond to the comment about suitable venues below.
>
> ### Optimization journal vs. NeurIPS
> We firmly believe that NeurIPS is an ideal venue for this work, as the original SVGD algorithm and analysis were published at NeurIPS [17], the bulk of all subsequent analyses were published at NeurIPS or ICML [16,9,14,20,25], and SVGD has since gained widespread use in the machine learning community in particular.  We thus believe that this work will be of the greatest interest and relevance to the NeurIPS community.

---

### Official Review · Reviewer_igw1 · 2023-07-04

**Soundness:** 3 good
**Presentation:** 4 excellent
**Contribution:** 4 excellent
**Rating:** 8
**Confidence:** 2

**Summary:**

The authors provide the first convergence guarantee for finite particle Stein Variational Gradient Descent (SVGD). Although I am not an expert on this topic, I believe this problem remained open for a long time, and it should be the first of many finite particle results to come.

While $(\log\log n)^{-1/2}$ is a fairly slow rate, I believe we should overlook the rate, but consider the significance of the technical leap taken by the authors to achieve a finite particle result at all. For this reason, I will recommend accept fro this paper.

**Strengths:**

1. This is the first finite particle convergence guarantee for SVGD.
2. The contents are well organized and presented.
3. The proof is concise and clean.

**Weaknesses:**

N/A

**Questions:**

Given that I am not an expert on this subject, I would like the authors to clarify a couple of questions for me.

1. What was the main conceptual challenge in establishing a finite particle guarantee, and how did this work overcome it? For me Theorem 3 reads like a bit of magic, and the desired result just appears without much intuition. I would like to understand how this came to be.

2. What do the authors perceive as the next challenge preventing improvements to this result? Similar to the previous question, I don't quite see on an intuitive level where the $log log n$ dependence came about, which the authors also believe can be improved. I would appreciate the authors can elaborate further on this topic.

---

> ### Author Rebuttal · Authors · 2023-08-10
>
> Thank you for the positive feedback. We are glad that you found our contribution significant, our work well presented, and our proof concise and clean. We provide responses to the detailed comments below.
>
> ### Main conceptual challenge in establishing a finite particle guarantee, and how this work overcomes it.
> SVGD is originally derived from finding descent directions of KL divergence between an approximation and the target distribution. Most prior convergence analyses [16,14,20] heavily rely on this KL-descent property, which ultimately yields a bound on KSD error but is only applicable to continuous SVGD because KL divergence is ill-defined between the n-particle discrete approximation and the continuous target distribution. Our work overcomes the difficulty, by explicitly controlling the 1-Wasserstein distance between the continuous and n-particle SVGD through a discretization error bound (a key challenge in deriving the explicit Theorem 1 was that the Wasserstein pseudo-Lipschitz constant of SVGD itself depends on the moment growth of the measures being compared, so our proof carefully tracks this growth in tandem with the discretization error) and, unlike [9], translating the resulting Wasserstein error into KSD error to enable its combination with convergence results for continuous SVGD.
>
> ### Intuition behind Theorem 3
> We will clarify that the aim of Theorem 3 is to combine Theorem 2 (the error of using n-particle SVGD to approximate continuous SVGD) and Corollary 1 (the error bound of continuous SVGD) into an error bound of n-particle SVGD. Since both bounds are formulated with the KSD metric, we used the triangle inequality of KSD to prove the result.
>
> ### The next challenge preventing improvements to this result; where the $\\log \\log n$ dependence came about
> Please see our response to questions shared by the reviewers.

---

> > ### Comment · Reviewer_igw1 · 2023-08-12
> > **Response**
> >
> > Thank you for the reply. I believe my questions are answered. While I am not an expert on this subject, and it's hard for me to justify raising the score, I would like to see this paper accepted given what I understand about it now.
> >
> > Therefore I will raise my score to 8, mostly in context of other reviews that are too pessimistic in my opinion and for non-technical reasons. I hope the AC will consider evaluating this work on a more fair benchmark, especially given this is the first finite particle guarantee.

---

### Official Review · Reviewer_g2wP · 2023-07-08

**Soundness:** 4 excellent
**Presentation:** 3 good
**Contribution:** 3 good
**Rating:** 6
**Confidence:** 3

**Summary:**

This work studies the non-asymptotic convergence rate of Stein variational gradient descent (SVGD), an algorithm for approximating a target probability distribution with a collection of particles. This work presents a finite-particle convergence rate for SVGD, which provides a measure of how quickly the algorithm converges to the target distribution with a finite number of particles. The convergence rate formula drives the kernel Stein discrepancy to zero at an order 1/√log log n rate, but the authors suspect that the dependence on n can be improved and hope that their proof strategy will serve as a template for future refinements.


**Strengths:**

This work provides the first finite-particle convergence rate for Stein variational gradient descent (SVGD), which is a popular algorithm for approximating a probability distribution with a collection of particles. This work presents an explicit, non-asymptotic proof strategy for the convergence rate formula --- which drives the kernel Stein discrepancy to zero at an order $1/\sqrt{\log \log n}$ rate providing a measure of how quickly the algorithm converges to the target distribution with a finite $n$ particles. Prior to this work, relatively little is known about SVGD's non-asymptotic approximation quality, despite that SVGD has demonstrated promising results for various inferential tasks. The authors also claims that it serves as a template for future refinements to improve the dependence on $n$. I highly believe in this, due to the soundness of technical tools the work adopted.

The authors provide a thorough discussion of the assumptions and conditions required for the convergence rate formula to hold, which helps to clarify the limitations and applicability of the formula. Finally, this work also includes a comprehensive list of references to related work, which provides a useful starting point for further research on SVGD and related algorithms.

**Weaknesses:**

Despite its success in providing the first non-asymptotic convergence rate, this work assumes a level of familiarity with the mathematical concepts and notation for experts, which may make it difficult for readers without a strong background in probability theory and optimization to follow. In addition, this work does not provide any experimental results or comparisons with other algorithms to demonstrate the practical usefulness of the convergence rate formula, and does not provide any specific information on how the dependence on $n$ can be improved, which may limit its usefulness for researchers looking to optimize the performance of SVGD. Lastly, this work focuses exclusively on SVGD and does not provide any insights or comparisons with other algorithms for approximating probability distributions, which may limit its broader relevance to the field.

**Questions:**

I am at an educative level but quite enjoy reading on this topic. I quite like the generic / infinite-particle continuum manner SVGD Algorithm 2 is written instead of the $n$-particle SVGD Algorithm 1. I wondered if this continuous-time approximation is the critical reason for the success of the non-asymptotic rate first established by the authors.

**Limitations:**

This paper contributes as a theoretical work and does not raise negative social impacts.

---

> ### Author Rebuttal · Authors · 2023-08-10
>
> We thank the reviewer for the positive feedback. We are pleased that the reviewer found our technical tools sound, our discussion of assumptions thorough, and also shared our vision for future refinements. Below is our detailed response:
>
> ### Assumed familiarity of optimization/probability theory concepts and notations
> Thank you for this feedback. We endeavor to make the final version maximally accessible, with all notation and concepts defined in Section 2. If there are particular concepts that the reviewer finds unclear, please let us know!
>
> ### Experimental results and comparisons with other algorithms
> The main focus of our work was not to propose a new algorithm or even to advocate for SVGD over alternative algorithms but rather to address the longstanding open question of whether a  unified convergence bound for finite-particle SVGD could be derived. As such, we believe that an experimental comparison with other algorithms would be out of scope in this work. There are however many empirical comparisons of SVGD with other algorithms in the cited literature, and, in the introduction, we do compare our theoretical result with prior work (including Liu [16], Gorham et al. [9], Korba et al. [14], and Salim et al. [20]) that studies the convergence of SVGD without providing unified convergence bounds or rates.
>
> ### How the dependence on n can be improved
> Please see our response to questions shared by the reviewers.
>
> ### This work focuses only on SVGD.
>
> Thank you for highlighting this opportunity to further contextualize our work. While the focus of this work is wholly on solving the open problem of establishing any unified convergence bound for finite-particle SVGD and we believe this initial rate will be improved in the future, we will better contextualize the initial rate and highlight what may be achievable in the future by comparing to other algorithms for approximating probability distributions. For example, MCMC methods like the unadjusted Langevin algorithm admit a much faster rate (see, e.g., the polynomial rate bounds of Balasubramanian et al. (2022)). However, as agreed upon by the reviewer and other reviewers (igw1,awG8), this work is still a “significant technical leap” to achieve the first non-asymptotic convergence rate for SVGD. We will also discuss a promising follow-up to this work by an independent research group (for anonymity reasons we omit the citation here) that has already begun investigating improved rates by modifying the SVGD algorithm.
>
> Our analysis also provides a template for studying convergence rates for SVGD-like algorithms. For example, Shi et al. (2021) proposed SVGD-like methods for sampling in constrained domains. However, their convergence analysis lacks a unified bound and rate. We could apply similar proof techniques in this submission to obtain a convergence rate for their algorithm.
>
> Reference:
> * Balasubramanian, K., Chewi, S., Erdogdu, M. A., Salim, A., & Zhang, S. (2022). Towards a theory of non-log-concave sampling: first-order stationarity guarantees for langevin monte carlo. In Conference on Learning Theory (pp. 2896-2923).
> * Shi, J., Liu, C., & Mackey, L. (2021). Sampling with mirrored Stein operators. arXiv preprint arXiv:2106.12506.
>
> ### The role of infinite-particle continuum manner SVGD
> We appreciate your positive response to how we present the continuous SVGD. The formulation in Algorithm 2 is indeed critical to our analysis. Our proof relies on an error bound for the continuous SVGD established by Corollary 1 and uses a discretization error bound (Theorem 2) to relate the continuous SVGD to n-particle SVGD.

---

> > ### Comment · Reviewer_g2wP · 2023-08-17
> >
> > Thank you for your informative response, especially the clarification of infinite-particle continuum limit. Indeed I believe this work should not be made obsolete due to its providing the first non-asymptotic convergence rate "as a significant leap". I have raised my score from 5 to 6 accordingly.

---

### Author Rebuttal · Authors · 2023-08-10

# Response to questions shared by Reviewer g2wP, igw1, and awG8:

### Source of $n$ dependence, potential avenues for rate improvement, and challenges involved.
The unified error bound of Theorem 3 reveals that the dependence on n arises from the tradeoff between the KSD discretization error bound ($a_{t-1}$), which grows double exponentially as the step size sum $b_{t-1}$ increases, and the infinite-particle SVGD error, which decreases proportionally to $\sqrt{b_{t-1}}$. The $\log\log n$ dependence is mainly caused by the double exponential growth of $a_{t-1}$, which can be traced back to Theorem 1. A key challenge in deriving the explicit Theorem 1 was that the Wasserstein pseudo-Lipschitz constant of SVGD itself depends on the moment growth of the measures being compared, so our proof carefully tracks this growth in tandem with the discretization error.  Any improvement in this discretization error bound growth rate would translate immediately into an improved approximation error rate for SVGD.  Moreover, one may be able to derive tighter bounds by analyzing the discretization error of KSD directly or by using an alternative intermediate metric in place of the 1-Wasserstein distance; finding the right metric that simultaneously remains small across the SVGD trajectory and is tractable to analyze is the main challenge.

Alternatively, instead of measuring convergence to an arbitrary sub-Gaussian target with respect to a large non-parametric measure like KSD, one could focus on the convergence of a more restricted set of moments (like means and variances) or a more restricted set of targets.  Our submission has already stimulated promising follow-up work (we have withheld the title and reference to preserve anonymity, but please let us know if we should reveal it) from an independent research group demonstrating rate improvements when the function class is more restricted and the target is Gaussian or strongly log concave.  Since the posting of our preprint on arXiv, another independent research group has built upon our work to show that a variant of SVGD converges at a much faster O(1/poly(n)) rate.  We will highlight these avenues for improvement and cite these follow-up works in the revision.

---

### Decision · Program_Chairs · 2023-09-21

**Decision:**

Accept (poster)

**Comment:**

Most of the reviewers deem the paper interesting. Please revise the paper and make it more readable for the machine learning audience.